# Large Graph Property Prediction via Graph Segment Training

**Kaidi Cao**[1]*, **Phitchaya Mangpo Phothilimthana**[2], **Sami Abu-El-Haija**[2], **Dustin Zelle**[2],
**Yanqi Zhou**[2], **Charith Mendis**[3]†, **Jure Leskovec**[1], **Bryan Perozzi**[2]
[1]Stanford University, [2]Google, [3]UIUC

## Abstract

Learning to predict properties of a large graph is challenging because each prediction requires the knowledge of an entire graph, while the amount of memory available during training is bounded. Here we propose Graph Segment Training (GST), a general framework that utilizes a divide-and-conquer approach to allow learning large graph property prediction with a constant memory footprint. GST first divides a large graph into segments and then backpropagates through only a few segments sampled per training iteration. We refine the GST paradigm by introducing a historical embedding table to efficiently obtain embeddings for segments not sampled for backpropagation. To mitigate the staleness of historical embeddings, we design two novel techniques. First, we finetune the prediction head to fix the input distribution shift. Second, we introduce *Stale Embedding Dropout* to drop some stale embeddings during training to reduce bias. We evaluate our complete method GST+EFD (with all the techniques together) on two large graph property prediction benchmarks: MalNet and TpuGraphs. Our experiments show that GST+EFD is both memory-efficient and fast, while offering a slight boost on test accuracy over a typical full graph training regime.

## 1 Introduction

Graph property prediction is a task of predicting a certain property or a characteristic of an entire graph [3]. Important applications include, predicting properties of molecules [32, 12], predicting properties of programs/code [1, 12, 6, 41] and predicting properties of organisms based on their protein-protein interaction networks [27, 42].

These popular graph property prediction tasks deal with relatively small graphs, so the scalability issue arises only from a large number of (small) graphs.

However, graph property prediction tasks also face another scalability challenge, which arises due to the large size of each individual graph, as some graphs can have millions or even billions of nodes and edges [10]. Training typical Graph Neural Networks (GNNs) to classify such large graphs can be computationally infeasible, as the memory needed scales at least linearly with the size of the graph [40]. This presents a challenge as even most powerful GPUs, which are optimized for handling large amounts of data, only have a limited amount of memory available.

Previous efforts to improve scalability of GNNs have mostly focused on developing methods for node-level and link-level prediction tasks, which can be performed using sampled subgraphs [11, 5, 13, 38, 43, 2, 21]. However, there is a lack of research on how to train scalable models for *property prediction of large graphs*. Training a model on a sampled subgraph alone is insufficient for these

---

*The work was partially completed during Kaidi Cao's internship at Google.
†The work was partially completed when Charith Mendis was a visiting researcher at Google.

37th Conference on Neural Information Processing Systems (NeurIPS 2023).

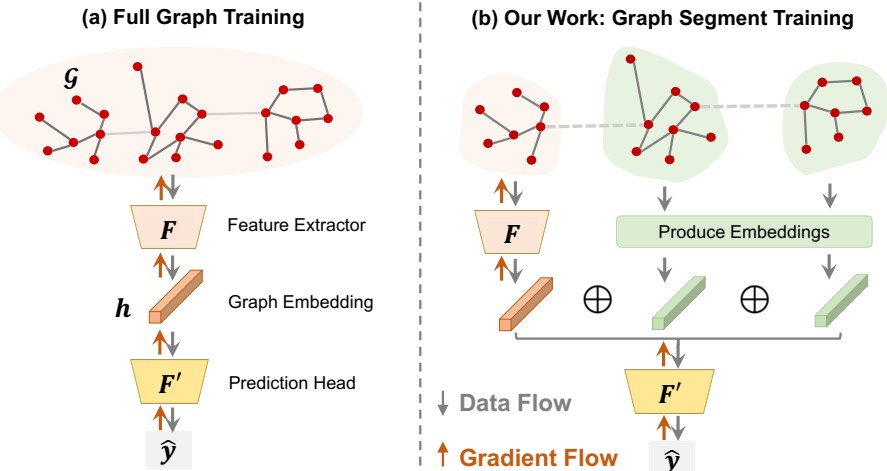

Figure 1: (a) **Full Graph Training**: Classically, models are trained using the entire graph, meaning all nodes and edges of the graph are used to compute gradients. For large graphs, this might be computationally infeasible. (b) **Graph Segment Training**: Our solution is to partition each large graph into smaller segments and select a random subset of segments to update the model; embeddings for the remaining segments are produced without saving their intermediate activations. The embeddings of all segments are combined to generate an embedding for the original large graph, which is then used for prediction. The important benefit is that GPU memory requirement only depends on the segment size (but not the full graph size).

types of tasks, as the subgraph sampled may not contain all the necessary information to make accurate predictions for the entire graph. For example, if the task is to predict the diameter of the graph, it is unlikely that a fixed-size subgraph would contain sufficient features for the GNN to make an accurate and reliable prediction. Thus, it is essential to aggregate information from the entire graph to predict a graph property.

In this paper, we address the problem of property prediction of large graphs. We propose Graph Segment Training (GST)[3], which is able to train on large graphs with constant (GPU) memory footprint.

Our approach partitions each large graph into smaller segments with a controlled size in the pre-processing phase. During the training process, a random subset of segments is selected to update the model at each step, rather than using the entire graph. This way, we need to maintain intermediate activations for only a few segments for backpropagation; embeddings for the remaining segments are created without saving their intermediate activations. The embeddings of all segments are then combined to generate an embedding for the original large graph, which is used for prediction. Therefore, each large graph has an upper bound on memory consumption during training regardless of its original size. This allows us to train the model on large graphs without running into an out-of-memory (OOM) issue, even with limited computational resources.

To accelerate the training process further, we introduce a historical embedding table to efficiently produce embeddings for graph segments that do not require gradients, as historical embeddings eliminate additional computation on such segments. However, the historical embeddings induce staleness issues during training, so we design two techniques to mitigate such issues in practice. First, we characterize the input distribution mismatch issue between training and test stages of the prediction head, and propose to finetune only the prediction head at the end of training to close the input distribution gap. Second, we identify bias in the loss function due to stale historical embeddings, and introduce *Stale Embedding Dropout* to drop some stale embeddings during training to reduce this bias. Our final proposed method, called GST+EFD, is both memory-efficient and fast.

We evaluate our method on the following datasets: MalNet-Tiny, MalNet-Large and TpuGraphs. A typical full graph training pipeline (Full Graph Training) can only train on MalNet-Tiny, and

---

[3]Source code available at `https://github.com/kaidic/GST`.

unavoidably reaches OOM on MalNet-Large and TpuGraphs dataset. On the contrary, we empirically show that the proposed GST framework successfully trains on arbitrarily large graphs using a single NVIDIA-V100 GPU with only 16GB of memory for MalNet-Large and four GPUs for TpuGraphs dataset, while maintaining comparable performance with Full Graph Training. We finally demonstrate that our complete method GST+EFD slightly outperforms GST by another 1-2% in terms of final evaluation metric, and simultaneously being 3x faster in terms of training time.

## 2 Preliminaries

**Notation.** For a function $f(\boldsymbol{a}) : \mathbb{R}^{d_0} \to \mathbb{R}^{d_1}$, we use $D_{\boldsymbol{a}}^k f[\boldsymbol{a}_0] \in \mathbb{R}^{d_0 \times d_1}$ to denote its $k$-th derivative of $f$ with respect to $\boldsymbol{a}$ evaluated at value $\boldsymbol{a}_0$. Let $f \circ g$ to denote the composition of function $f$ with $g$. We use $\odot$ to denote entry-wise product, and let $\boldsymbol{a}^{\odot 2} \triangleq \boldsymbol{a} \odot \boldsymbol{a}$. We use $\bigoplus$ to represent aggregation. $\bigoplus_{j \leq J} \boldsymbol{a}_j$ means aggregating the set $\{\boldsymbol{a}_j\}_{j \leq J}$, where $j$ indexes segments, and $J$ is the number of segments in a graph. We usually drop $j$ subscript from $\bigoplus$ for brevity. $\boldsymbol{a}_i \bigoplus \boldsymbol{b}_j$ aggregates both sets $\{\boldsymbol{a}_i\}$ and $\{\boldsymbol{b}_j\}$ together. $\bigoplus$ can be mean or sum operators when applying to vectors. We define the input graph as $\mathcal{G} = \{\mathcal{V}, \mathcal{E}\}$, where $\mathcal{V} = \{v_1, ..., v_m\}$ is the node set and $\mathcal{E} \subseteq \mathcal{V} \times \mathcal{V}$ is the edge set.

Let dataset $\mathcal{D} = \{(\mathcal{G}^{(i)}, y^{(i)})\}_{i \leq n}$ contain $n$ examples: each label $y^{(i)}$ is associated with $\mathcal{G}^{(i)}$.

**Graph Neural Network.** We consider a backbone graph neural network $F$ that takes a graph $\mathcal{G}^{(i)}$ and generates graph embedding $\boldsymbol{h}^{(i)} \in \mathbb{R}^{d_h}$, followed by a final prediction head $F'$ that takes graph embedding $\boldsymbol{h}^{(i)}$ and outputs final predictions: $\widehat{y}^{(i)} = (F' \circ F)(\mathcal{G}^{(i)})$. We optimize GNN with the loss function as $\mathcal{L}((F' \circ F)(\mathcal{G}^{(i)}), y^{(i)})$.

**Historical Embedding Table.** We define an embedding table $\mathcal{T} : \mathcal{K} \to \mathbb{R}^{d_h}$, where key-space $\mathcal{K} \subset \mathbb{Z} \times \mathbb{Z}$ is a tuple: (graph index $i \leq n$, segment index $j \leq J$). We use $\tilde{\boldsymbol{h}}_j^{(i)} = \mathcal{T}(i, j)$ to denote embedding for graph segment $\mathcal{G}_j^{(i)}$: an embedding not up to date with the current backbone $F$.

## 3 Our Method: GST+EFD

### 3.1 Graph Segment Training (GST)

Given a training graph dataset $\mathcal{D}_{\text{train}} = \{(\mathcal{G}^{(i)}, y^{(i)})\}_{i=1}^n$, a common SGD update step requires calculating the gradient:

$$\nabla_\theta \sum_{(\mathcal{G}^{(i)}, y^{(i)}) \in \mathcal{B}} \mathcal{L}((F' \circ F)(\mathcal{G}^{(i)}), y^{(i)})$$

where $\theta$ is trainable weights in $F' \circ F$, and $\mathcal{B}$ is a sampled minibatch. Graphs can differ in size (the number of nodes $|\mathcal{V}^{(i)}|$), with some being too large to fit into the device's memory. This is because the memory required to store all intermediate activations for computing gradients is proportional to the number of nodes and edges in the graph.

To address the above issue, we propose to partition each original input graph into a collection of graph segments, $i.e.$,

$$\mathcal{G}^{(i)} \approx \bigoplus \mathcal{G}_j^{(i)} \quad \text{for } j \in \{1, 2, \ldots, J^{(i)}\}$$

An example of a partition algorithm is METIS [16]. This preprocessing step will result in a training set $\mathcal{D}_{\text{train}} = \{(\bigoplus_{j \leq J^{(i)}} \mathcal{G}_j^{(i)}, y^{(i)})\}_{i=1}^n$. Number of partitions $J^{(i)}$ can vary across graphs, but the size of each graph segment can be bounded by a controlled size ($|\mathcal{V}_j^{(i)}| < m_{\text{GST}}, \forall (i, j)$) so that a batch of a fixed number of graph segments can always fit within the device's memory.

When processing graph segment $\mathcal{G}_j^{(i)}$, we can obtain its segment embedding through the backbone: $\boldsymbol{h}_j^{(i)} = F(\mathcal{G}_j^{(i)})$. The prediction head $F'$ requires information from the whole graph to make the prediction, thus we propose to aggregate all segment embeddings to recover the full graph embedding: $\boldsymbol{h}^{(i)} = \bigoplus \boldsymbol{h}_j^{(i)}$. A simple realization of this aggregation is mean pooling. Note that naïvely applying

the prediction head $F'$ on the aggregated graph embedding — $\widehat{y}^{(i)} = F'(\bigoplus h_j^{(i)})$ — would not provide any reduction in peak memory consumption, as we need to keep track of the activations of all graph segments $\{\mathcal{G}_j^{(i)}\}_{j \in J^{(i)}}$ to perform backpropagation.

Thus, we propose to perform backpropagation on only a few randomly sampled graph segments $\mathcal{S}^{(i)} \subseteq \{1, \ldots, J^{(i)}\}$ and generate embeddings without requiring gradients for the rest. We hereby denote $\boldsymbol{h}_s$ and $\bar{\boldsymbol{h}}_j$ as segment embeddings that require and do not require gradient, respectively. An entire graph embedding is then: $\boldsymbol{h}^{(i)} \approx \{\boldsymbol{h}_s^{(i)}\}_{s \in \mathcal{S}^{(i)}} \bigoplus \{\bar{\boldsymbol{h}}_j^{(i)}\}_{j \notin \mathcal{S}^{(i)}} \triangleq \boldsymbol{h}_s^{(i)} \bigoplus \bar{\boldsymbol{h}}_j^{(i)}$. We name this general pipeline as GST and summarize it in Algorithm 1.

---

**Algorithm 1** General Framework of GST

---

**Require:** A preprocessed training graph dataset $\mathcal{D}_{\text{train}} = \{(\bigoplus \mathcal{G}_j^{(i)}, y^{(i)})\}_{i=1}^n$. A parameterized backbone $F$ and a prediction head $F'$.
1: **for** $t = 1$ to $T_0$ **do**
2:     $\mathcal{B} \leftarrow \text{SampleMiniBatch}(\mathcal{D}_{\text{train}})$
3:     **for** $(\mathcal{G}^{(i)}, \widehat{y}^{(i)})$ in $\mathcal{B}$ **do**
4:         $\{\mathcal{G}_s^{(i)}\}_{s \in \mathcal{S}^{(i)}} \leftarrow \text{SampleGraphSegments}(\mathcal{G}^{(i)})$
5:         $\bar{\boldsymbol{h}}_j^{(i)} \leftarrow \text{ProduceEmbedding}(\mathcal{G}_j^{(i)})$ for $j \notin \mathcal{S}^{(i)}$
6:         $\boldsymbol{h}_s^{(i)} \leftarrow F(\mathcal{G}_s^{(i)})$    **for** $s \in \mathcal{S}^{(i)}$
7:     **end for**
8:     SGD on loss $\leftarrow \frac{1}{|\mathcal{B}|} \sum_i \mathcal{L}\left(F'(\boldsymbol{h}_s^{(i)} \bigoplus \bar{\boldsymbol{h}}_j^{(i)}), \widehat{y}^{(i)}\right)$
9: **end for**

---

One implementation of ProduceEmbedding($\cdot$) in Algorithm 1 is to use the same feature encoder $F$ to forward all the segments in $\{\mathcal{G}_j^{(i)}\}_{j \notin \mathcal{S}^{(i)}}$ without storing any intermediate activation (by stopping gradient).

## 3.2 GST with Historical Embedding Table

Calculating $\bar{\boldsymbol{h}}_j$ by stopping gradient guarantees an upper bound on peak memory consumption. However, since we do not need gradients for segments $\{\mathcal{G}_j^{(i)}\}_{j \notin \mathcal{S}^{(i)}}$, computing forward pass on these segments can be avoided to make training faster. To achieve this, we use historical embeddings acquired in previous training iterations $\tilde{\boldsymbol{h}}_j^{(i)} = \mathcal{T}(i, j)$. With an embedding table $\mathcal{T}$, one can implement ProduceEmbedding($\cdot$) by fetching the corresponding embedding from the table without any computation. We update the embedding table after conducting the forward pass on a graph segment. We optimize the following loss $\mathcal{L}(F'(\boldsymbol{h}_s^{(i)} \bigoplus \tilde{\boldsymbol{h}}_j^{(i)}), y^{(i)})$ during training. We name the embedding version of our algorithm as GST+E.

Note that GST+E has runtime advantages over GST. For each segment $\mathcal{G}_j^{(i)}$ that does not require gradient, GST needs to run an actual forward pass over $\mathcal{G}_j^{(i)}$, while GST+E only needs to fetch the embedding from a hash table. GST+E has a small overhead from writing the updated embedding of $\mathcal{G}_s^{(i)}$ back into the table $\mathcal{T}$, which is relatively quick and can be run in a separate thread so that it does not impede the main training algorithm until the next iteration.

The drawback of GST+E is that historical embeddings from $\mathcal{T}$ may be stale;

$\tilde{\boldsymbol{h}}_j^{(i)}$ can be the result of an out-dated feature extractor $F$. This type of staleness can hurt the training in various ways. Below, we provide two techniques to mitigate the staleness issue.

## 3.3 Prediction Head Finetuning

Let's compare the input-output distribution of the prediction head $F'$ during the training and inference stage. We have training distribution $\mathcal{P}_{\text{train}}(\boldsymbol{h}, y) = \mathcal{P}(\boldsymbol{h}_s \bigoplus \tilde{\boldsymbol{h}}_j, y)$ and test distribution $\mathcal{P}_{\text{test}}(\boldsymbol{h}, y) = \mathcal{P}(\bigoplus \boldsymbol{h}_j, y)$. Regardless of the innate distribution shift between the training and test stage of the dataset, we note that stale historical embeddings can further widen the gap between the training and

test distributions. In this case, the minimizer of the expected training loss does not minimize the expected test loss:

$$\arg\min_{\theta} \mathbb{E}_{(\boldsymbol{h},y)\sim\mathcal{P}(\boldsymbol{h}_s\bigoplus\tilde{\boldsymbol{h}}_j,y)}\mathcal{L}(F'(\boldsymbol{h}),y) \neq \arg\min_{\theta} \mathbb{E}_{(\boldsymbol{h},y)\sim\mathcal{P}(\bigoplus\boldsymbol{h}_j,y)}\mathcal{L}(F'(\boldsymbol{h}),y)$$

To mitigate the distribution misalignment, we introduce the Prediction Head Finetuning technique. Concretely, at the end of training, we update each embedding $\boldsymbol{h}_j^{(i)}$ in the embedding table $\mathcal{T}$ by forwarding each graph segment in the training set with the most current feature encoder $F$. We then finetune only the prediction head $F'$ with all the input embeddings up-to-date. We use GST+EF to denote GST+E refined with the Prediction Head Finetuning technique.

The overhead from the finetuning is minimal, as we need to update the embedding table $\mathcal{T}$ only once. The rest of the finetuning stage does not involve the notoriously slow graph convolution because the prediction head $F'$ is simply a multi-layer perceptron.

### 3.4 Stale Embedding Dropout

The finetuning technique primarily addresses the negative impact of stale embeddings on prediction head $F'$. However, the staleness also impacts the backbone $F$. Prior works studying the effects of stale historical embeddings commonly assume that historical embeddings do not become too stale, *i.e.*, $\|\tilde{\boldsymbol{h}}_j^{(i)} - \bar{\boldsymbol{h}}_j^{(i)}\| \leq \epsilon, \forall(i,j)$. Given this assumption, if the neural network $(F' \circ F)(\cdot)$ is $k$-Lipschitz continuous, the gradients will also be bounded and never run too far from its true estimation, *i.e.*, $\|\nabla F'(\tilde{\boldsymbol{h}}_j^{(i)}) - \nabla F'(\bar{\boldsymbol{h}}_j^{(i)})\| \leq k' \cdot \epsilon$. Thus, the network can often converge to the similar local minima even when using historical embeddings.

The above assumption does not hold under our GST+E framework. The rationale is that $\tilde{\boldsymbol{h}}_j^{(i)}$ gets updated infrequently in the embedding table $\mathcal{T}$. Suppose we iterate through every graph $\mathcal{G}^{(i)}$ in the training set $\mathcal{D}_{\text{train}}$ for each epoch, every time we train on a graph $\mathcal{G}^{(i)} \approx \bigoplus \mathcal{G}_j^{(i)}$, only a few graph segments $\mathcal{G}_s^{(i)}$ will be updated in the table $\mathcal{T}$. This implies that all the other segment embeddings of $\mathcal{G}^{(i)}$ will be at least $n$-iteration stale, with $n$ being the number of graphs in the training set, and the most outdated segment embedding could be approximately $nJ^{(i)}/S^{(i)}$-iteration stale, where $S^{(i)}$ is $|\mathcal{S}^{(i)}|$.

This staleness introduces an additional source of bias and variance to the stochastic optimization; the loss function calculated with historical embeddings is no longer an unbiased estimation of its true value. To mitigate the negative impact of historical embeddings on loss function estimation, we propose the second technique, *Stale Embedding Dropout* (SED). Unlike a standard Dropout, which uniformly drops elements and weighs up the rest, we propose to drop only stale segment embeddings and weigh up only segment embeddings that are up-to-date. Concretely, assume with the keep probability $p$, the weight $\eta$ for each segment is defined as:

$$\eta^{(i)} = \begin{cases} p + (1-p)\frac{J^{(i)}}{S^{(i)}} & \text{for } \mathcal{G}_s^{(i)} \\ 0 & \text{for } \mathcal{G}_j^{(i)}, \text{ with prob. } (1-p) \\ 1 & \text{for } \mathcal{G}_j^{(i)}, \text{ with prob. } p \end{cases} \tag{1}$$

Please refer to the theoretical analysis in the next section. By combining the two proposed techniques, we denote our final algorithm as GST+EFD, which we summarized in Algorithm 2 in Appendix B.

## 4 Theoretical Analysis

We characterize the effect of stale historical embeddings by studying the difference between $\mathcal{L}(F'(\boldsymbol{h}_s^{(i)}\bigoplus\tilde{\boldsymbol{h}}_j^{(i)}))$ and $\mathcal{L}(F'(\bigoplus\boldsymbol{h}_j^{(i)}))$. Assume $\mathcal{G}^{(i)}$ has $J^{(i)}$ segments and we perform back-propagation on $S^{(i)}$ segments. We let $\delta^{(i)} \triangleq \boldsymbol{h}_s^{(i)}\bigoplus\tilde{\boldsymbol{h}}_j^{(i)} - \bigoplus\boldsymbol{h}_j^{(i)}$ be the perturbation on the graph embedding.

We apply Taylor expansion around $\delta^{(i)} = 0$ on the final loss to analyze the effect of this perturbation.

$$\mathbb{E}_s \mathcal{L}(F'(\boldsymbol{h}_s^{(i)} \bigoplus \tilde{\boldsymbol{h}}_j^{(i)})) - \mathcal{L}(F'(\bigoplus \boldsymbol{h}_j^{(i)})) \tag{2}$$

$$=\mathbb{E}_s \mathcal{L}(F'(\bigoplus \boldsymbol{h}_j^{(i)} + \delta^{(i)})) - \mathcal{L}(F'(\bigoplus \boldsymbol{h}_j^{(i)}))$$

$$\approx \sum_j \mathbb{E}_{\delta_j^{(i)}} \underbrace{D_{\boldsymbol{h}_j^{(i)}}(\mathcal{L} \circ F')[\boldsymbol{h}_j^{(i)}]\delta_j^{(i)}}_{B} + \underbrace{\frac{1}{2}\delta_j^{(i)\top}(D_{\boldsymbol{h}_j^{(i)}}^2(\mathcal{L} \circ F')[\boldsymbol{h}_j^{(i)}])\delta_j^{(i)}}_{R}$$

In the equation above, the first-order term acts as a bias term introduced by the stale historical embedding, and the second-order term acts as an additional regularizer.

Let ET denote using the embedding table without applying SED. We analyze the effect of ET and SED under the Taylor Expansion in Eq. 2 by substituting different $\delta^{(i)}$. For the first term, we have

$$\mathbb{E}_{\delta_j^{(i)\mathrm{ET}}}[B] = C \times \frac{J^{(i)} - S^{(i)}}{J^{(i)}}\mathbb{E}(\tilde{\boldsymbol{h}}_j^{(i)} - \boldsymbol{h}_j^{(i)})$$

$$\mathbb{E}_{\delta_j^{(i)\mathrm{SED}}}[B] = C \times \frac{J^{(i)} - S^{(i)}}{J^{(i)}}\mathbb{E}(\tilde{\boldsymbol{h}}_j^{(i)} - \boldsymbol{h}_j^{(i)})p$$

where $C$ is a constant matrix.

We extend the above analysis to the following theorem. Please find the complete proof in Appendix A.

**Theorem 4.1.** *Under proper conditions, SED with a keep ratio $p$ ensures to reduce bias term introduced by historical embeddings by a factor of $p$, while introducing another regularization term.*

Theorem 4.1 indicates that SED can reduce the bias in the loss function introduced by the stale historical embeddings, at the cost of another regularization term, which might potentially increase total regularization. Prior works commonly make a hidden assumption that $\mathbb{E}(\tilde{\boldsymbol{h}}_j^{(i)} - \boldsymbol{h}_j^{(i)})$ is so small that the negative effect may be neglected. In our setting, the historical embeddings can be very stale, so having a $p$ factor helps reduce bias. It is worthwhile to check the limiting cases. If $p = 1$ (keeping all the stale embeddings without dropping any), both $\mathbb{E}_{\delta_j^{(i)\mathrm{SED}}}[B]$ and $\mathbb{E}_{\delta_j^{(i)\mathrm{SED}}}[R]$ degrade to the result of ET. If $p = 0$ (droping all the stale embeddings), then the algorithm degrades to training on only $S^{(i)}$ segments without aggregating other segments (which we denote as GST-One, when $S^{(i)} = 1$). $\mathbb{E}_{\delta_j^{(i)\mathrm{SED}}}[B] = 0$ indicates that there is no stale bias in this case. However, the term $\mathbb{E}_{\delta_j^{(i)\mathrm{SED}}}[R]$ could become too large so that it impedes training.

## 5 Experiments

### 5.1 Experimental Setup

**Datasets.** MalNet [10] is a large-scale graph representation learning dataset, with the goal to predict the category of a function call graph. MalNet is the largest public graph database constructed to date in terms of average graph size. Its widely-used split is called *MalNet-Tiny*, containing 5,000 graphs across balanced 5 types, with each graph containing at most 5,000 nodes. To evaluate our approach on the regime where the graph size is large, we construct an alternative split from the original MalNet dataset, which we named *MalNet-Large*. *MalNet-Large* also contains 5,000 graphs across balanced 5 types. *MalNet-Large*'s average graph size reaches 47k with the largest graph containing 541k nodes. We will release our experimental split for *MalNet-Large* to promote future research.

TpuGraphs is an internal large scale graph regression dataset, whose goal is to predict an execution time of an XLA's HLO graph with a specific compiler configuration on a Tensor Processing Unit (TPU). XLA [28] is a production backend compiler for various machine learning frameworks, including TensorFlow, PyTorch, and JAX. In this particular dataset, the compiler configuration controls physical layouts of tensors in the graph, and the runtime is measured on TPU v3 [15]. This dataset cares more about the ranking of the configurations for each graph than the absolute runtimes, since the ultimate goal is to use a model to select the best configuration for each graph.[4] TpuGraphs

---

[4]We have made public a dataset [24] that closely parallels our internal dataset.

Table 1: Test accuracy on MalNet-Tiny and MalNet-Large. We report the standard deviation over five runs. GST+EFD achieves better accuracy than Full Graph Training, and GST, while being much more memory efficient and computationally faster.

| Dataset | MalNet-Tiny | | | MalNet-Large | | |
| Backbone | GCN | SAGE | GraphGPS | GCN | SAGE | GraphGPS |
| --- | --- | --- | --- | --- | --- | --- |
| Full Graph Training | 87.84±1.37 | 88.08±1.68 | 90.82±0.59 | OOM | OOM | OOM |
| GST | 88.26±0.80 | 88.42±1.03 | 91.03±0.81 | 88.35±1.14 | 88.62±0.82 | 91.39±0.85 |
| GST-One | 71.62±3.85 | 72.64± 4.73 | 77.63±3.15 | 60.41±6.29 | 57.13±7.36 | 66.82±4.71 |
| GST+E | 86.53±1.18 | 86.82±0.93 | 89.75±0.89 | 48.42±6.61 | 43.28±7.01 | 62.47±3.19 |
| GST+EF | 87.67±0.78 | 87.83±0.81 | 90.52±0.71 | 84.83±0.96 | 85.26±0.87 | 91.33±0.65 |
| GST+ED | 88.18±0.48 | 88.50±0.74 | 90.96±0.68 | 82.17±4.74 | 71.83±6.31 | 89.46±1.36 |
| GST+EFD | 88.78±0.45 | 89.24±0.53 | 92.46± 0.66 | 89.67±0.71 | 89.78±0.68 | 92.52±0.58 |

is similar to the dataset used in [17], but the runtime prediction is at the entire graph level rather than at the kernel (subgraph) level. TpuGraphs contains 5,153 HLO graphs and a total of 757,375 unique pairs of graphs and configurations. The average graph size is 38k, and the maximum is 615k. From the perspective of GST, a graph together with a configuration defines one $\mathcal{G}^{(i)}$ because the configuration is featurized as parts of input node features to the GNN.

Please refer to Table 4 in Appendix for detailed statistics.

**Methods.** We test combinations of the following proposed techniques and some baselines. (1) Full Graph Training: we train on all graphs in their original scale without applying any partitioning beforehand. (2) GST-One: we partition the original graph into a collection of graph segments $\mathcal{G}^{(i)} \approx \bigoplus \mathcal{G}_j^{(i)}$, but we randomly select only one segment $\mathcal{G}_j^{(i)}$ for each graph to train every iteration. (3) GST: following the general GST framework described in Algorithm 1, we replace ProduceEmedding($\cdot$) by using the same feature encoder $F$ to forward all the segments in $\{\mathcal{G}_j^{(i)}\}_{j \notin \mathcal{S}^{()}}$ without storing any intermediate activation. We set $S^{(i)} = 1$ in our experiments. (4) E: we introduce an embedding table $\tilde{\boldsymbol{h}}_j^{(i)} = \mathcal{T}(i, j)$ to store the historical embedding of each graph segment, and we fetch the embedding from $\mathcal{T}$ if we do not need to calculate gradient for the corresponding segment. (5) F: in addition to introducing the embedding table $\mathcal{T}$, we finetune the prediction head $F'$ with all up-to-date segment embeddings at the end of training. (6) D: we apply SED defined in Eq. 1 during training.

When these techniques are combined, we concatenate the acronyms with a "+" to GST as an abbreviation. We conduct all the experiments on MalNet with a single NVIDIA-V100 GPU with 16GB of memory, and four NVIDIA-V100 GPUs (for data parallelism) with 16GB of memory for TpuGraphs. Please refer to Appendix B for additional implementation details.

## 5.2 Empirical Results on MalNet

To demonstrate the general applicability of our proposed GST framework, we consider three backbones, namely, GCN [19], SAGE [11], and GraphGPS [25]. GCN and SAGE are two popular GNN architectures. GraphGPS is a Graph Transformer that recently achieves state-of-the-art performance on many graph-level tasks, but is well-known for its issue on scalability. We report the top-1 test accuracy of various methods on MalNet-Tiny and MalNet-Large in Table 1. We include MalNet-Tiny in this study because its graphs are relatively small so that it is still possible to run Full Graph Training.

Notably, we observe that GST slightly outperforms Full Graph Training in terms of test accuracy on MalNet-Tiny. GST has exactly the same number of weight parameters with Full Graph Training. This implies that GST potentially has a better hierarchical graph pooling mechanism that leads to better generalization. As we step from MalNet-Tiny to MalNet-Large, Full Graph Training strategy can no longer fit the large graphs on a GPU, so we report OOM in the table. GST's estimation on graph segment embeddings $\bar{\boldsymbol{h}}_j^{(i)}$ that do not require gradients is accurate, and thus does not suffer from staleness issues. Therefore, we use GST as an estimation for the performance of Full Graph Training on MalNet-Large.

Naïvely training on only one graph segment (GST-One) yields inferior performance than Full Graph Training and GST, showing that it is essential to aggregate embeddings from all graph segments.

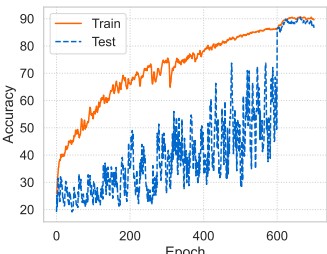 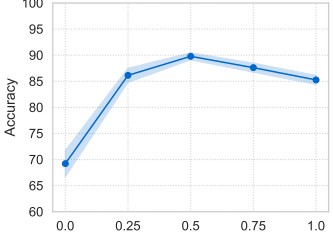 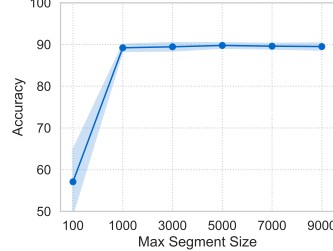

Figure 2: Accuracy curve on MalNet-Large of GST+EFD with SAGE backbone. We start Prediction Head Finetuning at epoch 600.

Figure 3: Ablation study on the keep ratio $p$ in SED. We report test accuracy of GST+EFD with SAGE backbone on MalNet-Large for 5 runs.

Figure 4: Ablation study on maximum segment size. We report test accuracy of GST+EFD with SAGE backbone on MalNet-Large for 5 runs.

Solely introducing the historical embedding table (GST+E) significantly deteriorates the optimization due to the staleness issue. Each of the proposed techniques (Prediction Head Finetuning and SED) individually is beneficial in combating the staleness issue. The combination of our two techniques (GST+EFD) achieves the best performance, slightly outperforming GST by another 1-2% in terms of final evaluation metric.

## 5.3 Empirical results on TpuGraphs dataset

Table 2: Training and test ordered pair accuracy (OPA) on TpuGraphs Dataset.

| | Train OPA | Test OPA |
|---|---|---|
| Full Graph Training | OOM | OOM |
| GST | 87.43 | 86.84 |
| GST-One | 67.19 | 81.03 |
| GST+E | 74.02 | 82.62 |
| GST+EFD | 71.40 | 89.20 |

Table 3: Runtime analysis (average training time per iteration in ms) on MalNet-Large dataset.

| | GCN | SAGE | GraphGPS |
|---|---|---|---|
| Full Graph Training | OOM | OOM | OOM |
| GST | 720.8 | 706.3 | 1285.7 |
| GST-One | 242.2 | 239.6 | 441.4 |
| GST+E | 258.4 | 253.6 | 451.9 |
| GST+EFD | 247.9 | 244.7 | 448.2 |

As mentioned in Section 5.1, we care more about the ranking of configurations for each graph than the absolute runtimes in this dataset. Thus, we report the ordered pair accuracy (OPA) averaged over all computational graphs in Table 2. OPA metric is defined as:

$$\text{OPA}(y, \widehat{y}) = \frac{\sum_i \sum_j \mathbb{I}[\widehat{y}_i > \widehat{y}_j] \cdot \mathbb{I}[y_i > y_j]}{\sum_i \sum_j \mathbb{I}[y_i > y_j]}$$

With some compiler domain knowledge, we found it better to predict each graph segment's runtime first and then aggregate them using sum pooling. This means the prediction head is part of $F$, and $F'$ is simply a summation function. Since there are no learnable weights in $F'$, we omit Prediction Head Finetuning in this experiment, and GST+EFD in Table 2 excludes the finetuning stage. We observe a clear tradeoff between fitting training examples and generalization. First, GST has a much higher training OPA than the other methods, indicating accurate estimation on segments that do not require gradient is essential for training OPA. Training with one segment only or with the embedding table yields lower training OPA, and consequently lower test OPA as well. SED in GST+EFD functions as a regularization technique. Although it slightly lowers the training OPA compared to GST+E, it achieves better test OPA, even than GST, due to bias mitigation.

## 5.4 Ablation Studies

**Effect of finetuning.** We visualize the training/test accuracy curve of GST+EFD over time in Figure 2. The staleness introduced by historical embeddings drastically hurts generalization, as shown for the first 600 epochs. We start finetuning at epoch 600, and the gap between training and test accuracy decreases by a large margin instantly.

**Ablation study on segment dropout ratio.** To analyze the effect of the keep ratio $p$ in SED, we vary its value from 0 to 1 and visualize the results in Figure 3. When $p = 1$, GST+EFD degrades back to using the historical embedding table without SED, as the performance decreases due to staleness. When $p = 0$, GST+EFD becomes GST-One, where we drop all the stale historical embeddings. This extreme case introduces too heavy regularization that impedes the model from fitting the training data, leading to a decrease in test performance ultimately. We found that $p = 0.5$ achieves a satisfactory tradeoff between fitting the training data and adding a proper amount of regularization.

**Ablation study on segment size.** We also alter the maximum segment size and visualize the results in Figure 4. A smaller maximum segment size will result in much more number of segments. Interestingly, we found that the proposed GST+EFD is very robust to the choice of the maximum segment size, as long as the segment size is reasonally large.

**Ablation study on partition algorithms.** Please refer to Appendix C.

## 5.5 Runtime Analysis

Next, we empirically compare runtime of different variants under the proposed GST framework. We summarize an average time for one forward-backward pass during training on MalNet-Large dataset in Table 3. Since GST runs inference for the graph segments that do not require gradients, the runtime of GST is significantly higher than others'. We also found that GST+E's and GST+EFD's runtime are very close to GST-One's; this means the overhead of fetching embeddings from the embedding table $\mathcal{T}$ is minimal. Moreover, GST+EFD's runtime is slightly lower than GST+E's because in the implementation, we can skip the fetching process if an embedding is set to be dropped. This result demonstrates that our proposed GST+EFD not only is efficient in terms of memory usage but also reduces training time significantly.

## 6  Related Works

**Graph property prediction.** In the context of graph property prediction, a model must predict a certain characteristic associated with the whole graph. Standard graph neural networks produce node embeddings as outputs [19, 11, 34]. To create a graph embedding (a vector representing the entire graph) out of the node embeddings, pooling methods are usually applied at the end. Common approaches to this problem include simply summing up or averaging all the node embeddings [8], or introducing a "virtual node" connected to all the other nodes [23]. Fully-connected Graph Transformer was recently proposed with an outstanding success on existing graph property prediction benchmarks [35, 25]. However, the fully-connected attention matrix limits the applicability of Graph Transformer to only small graphs with limited number of nodes [26].

**GNN with graph partitioning.** ClusterGCN [7] is designed for node-level tasks by training on graph segments. It partitions a graph into graph segments and randomly selects graph segments to form a minibatch during training. ROC [14], PipeGCN [29] and BNS-GCN [30] achieve distributed node-level GCN training through partitioning a graph into small segments such that each could be fitted into a single GPU memory, and then training multiple segments in parallel. All the above graph partitioning techniques for GNNs rely on the fact that an ego-subgraph (a subgraph centered around a node) contains sufficient information to make a prediction for the centered node. This is not true for graph property prediction tasks where we need to aggregate information from the whole graph to make an accurate prediction.

**GNN with historical embeddings.** The idea of historical embeddings was first introduced in VR-GCN [4], which uses historical embeddings to control the variance of neighborhood sampling. GNNAutoScale [9] incorporates historical embeddings to recover a more accurate neighborhood estimation in a scalable fashion. Developed upon GNNAutoScale, Yu et al. [37] uses a momentum step to incorporate historical embeddings when updating feature representations to further alleviate the staleness issue. These prior works maintain a historical embedding for each node because they consider node-level tasks. As we consider graph property prediction tasks, we record a historical embedding for each graph segment rather than each node.

# 7 Conclusion

We study how to train a GNN model for large graph property prediction tasks. We propose Graph Segment Training (GST), a general framework for learning large graph property prediction tasks with a constant memory footprint. We further introduce a historical embedding table to efficiently produce embeddings for graph segments that do not require gradients, and design two novel techniques — Prediction Head Finetuning and Stale Embedding Dropout — to mitigate the staleness issue. In conclusion, we believe that our proposed method is a step toward making graph property prediction learning more practical and scalable.

## Acknowledgements

KC and JL acknowledge the support of DARPA under Nos. HR00112190039 (TAMI), N660011924033 (MCS); ARO under Nos. W911NF-16-1-0342 (MURI), W911NF-16-1-0171 (DURIP); NSF under Nos. OAC-1835598 (CINES), OAC-1934578 (HDR), CCF-1918940 (Expeditions), NIH under No. 3U54HG010426-04S1 (HuBMAP), Stanford Data Science Initiative, Wu Tsai Neurosciences Institute, Amazon, Docomo, GSK, Hitachi, Intel, JPMorgan Chase, Juniper Networks, KDDI, NEC, and Toshiba. CM's contributions were partially supported by ACE, one of the seven centers in JUMP 2.0, a Semiconductor Research Corporation (SRC) program sponsored by DARPA and NSF under grant CCF-2316233. The content is solely the responsibility of the authors and does not necessarily represent the official views of the funding entities.

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

# A    Proofs and Derivations

**Theorem A.1.** *Under proper a condition that $W \cdot \tilde{h}_j^{(i)} \approx 0$ and $W \cdot h_j^{(i)} \approx 0$, where $W$ is the first linear transformation in $F'$, SED with a keep ratio $p$ ensures to reduce bias term introduced by historical embeddings by a factor of $p$, while introducing another regularization term.*

*Proof.* Let $\delta^{(i)} \triangleq h_s^{(i)} \bigoplus \tilde{h}_j^{(i)} - \bigoplus h_j^{(i)}$ be the perturbation on the graph embedding. We use ET to denote using the embedding table without applying SED. For GST+E, we have

$$\delta_j^{(i)\text{ET}} = \begin{cases} 0 & \text{with prob. } \frac{S^{(i)}}{J^{(i)}} \\ \tilde{h}_j^{(i)} - h_j^{(i)} & \text{with prob. } \frac{J^{(i)} - S^{(i)}}{J^{(i)}} \end{cases}$$

The randomness above comes from the fact that each segment $\mathcal{G}_j^{(i)}$ is selected for backpropagation with probability $\frac{S^{(i)}}{J^{(i)}}$.

For SED, there are two folds of randomness when training on graph $\mathcal{G}^{(i)}$: randomly selecting $S^{(i)}$ segments to train and randomly select stale embedding $\tilde{h}_j^{(i)}$ to drop. Thus we can rewrite $\delta_j^{(i)}$ as

$$\delta_j^{(i)\text{SED}} = \begin{cases} \frac{(1-p)(J^{(i)} - S^{(i)})}{S^{(i)}} h_j^{(i)} & \text{with prob. } \frac{S^{(i)}}{J^{(i)}} \\ -h_j^{(i)} & \text{with prob. } \frac{(1-p)(J^{(i)} - S^{(i)})}{J^{(i)}} \\ \tilde{h}_j^{(i)} - h_j^{(i)} & \text{with prob. } \frac{p(J^{(i)} - S^{(i)})}{J^{(i)}} \end{cases}$$

We apply Taylor expansion around $\delta_j^{(i)} = 0$ on the final loss to analyze the effect of this perturbation. In Section A.2 of Wei et al. [31], when $W \cdot \tilde{h}_j^{(i)} \approx 0$ and $W \cdot h_j^{(i)} \approx 0$, the perturbation of $\delta^{(i)}$ to the loss function might not be too large, so it supports the use of Taylor Expansion.

$$\mathcal{L}(F'(h_s^{(i)} \bigoplus \tilde{h}_j^{(i)})) - \mathcal{L}(F'(\bigoplus h_j^{(i)}))$$
$$= \mathcal{L}(F'(\bigoplus(h_j^{(i)} + \delta_j^{(i)}))) - \mathcal{L}(F'(\bigoplus h_j^{(i)}))$$
$$\approx \sum_j D_{h_j^{(i)}}(\mathcal{L} \circ F')[h_j^{(i)}]\delta_j^{(i)} + \frac{1}{2}\delta_j^{(i)\top}(D_{h_j^{(i)}}^2(\mathcal{L} \circ F')[h_j^{(i)}])\delta_j^{(i)}$$

Note that we randomly select segments with index $s$ during training, we can then derive an approximation of the expected difference during training as

$$\mathbb{E}_s \mathcal{L}(F'(h_s^{(i)} \bigoplus \tilde{h}_j^{(i)})) - \mathcal{L}(F'(\bigoplus h_j^{(i)})) \tag{3}$$
$$\approx \sum_j \mathbb{E}_{\delta_j^{(i)}} \underbrace{D_{h_j^{(i)}}(\mathcal{L} \circ F')[h_j^{(i)}]\delta_j^{(i)}}_{B} + \underbrace{\frac{1}{2}\delta_j^{(i)\top}(D_{h_j^{(i)}}^2(\mathcal{L} \circ F')[h_j^{(i)}])\delta_j^{(i)}}_{R}$$

We can then compare the effect of SED by substituting the two versions of $\delta_j^{(i)}$ into Eq. 3.

So for the first term, we have

$$\mathbb{E}_{\delta_j^{(i)\text{ET}}}[B] = \langle D_{h_j^{(i)}}(\mathcal{L} \circ F')[h_j^{(i)}], \mathbb{E}\delta_j^{(i)} \rangle$$

$$= \langle D_{h_j^{(i)}}(\mathcal{L} \circ F')[h_j^{(i)}], \frac{J^{(i)} - S^{(i)}}{J^{(i)}} \mathbb{E}(\tilde{h}_j^{(i)} - h_j^{(i)}) \rangle$$

$$\mathbb{E}_{\delta_j^{(i)\text{SED}}}[B] = \langle D_{h_j^{(i)}}(\mathcal{L} \circ F')[h_j^{(i)}], \mathbb{E}\delta_j^{(i)} \rangle$$

$$= \langle D_{h_j^{(i)}}(\mathcal{L} \circ F')[h_j^{(i)}], \frac{J^{(i)} - S^{(i)}}{J^{(i)}} \mathbb{E}(\tilde{h}_j^{(i)} - h_j^{(i)}) * p \rangle$$

whereas for the second term, we have

$$\mathbb{E}_{\delta_j^{(i)\text{ET}}}[R] = \langle D^2_{\boldsymbol{h}_j^{(i)}}(\mathcal{L} \circ F')[\boldsymbol{h}_j^{(i)}], \frac{\mathbb{E}\delta_j^{(i)}\delta_j^{(i)\top}}{2} \rangle$$

$$= \langle D^2_{\boldsymbol{h}_j^{(i)}}(\mathcal{L} \circ F')[\boldsymbol{h}_j^{(i)}], \frac{J^{(i)} - S^{(i)}}{2J^{(i)}}(\tilde{\boldsymbol{h}}_j^{(i)} - \boldsymbol{h}_j^{(i)})^{\odot 2} \rangle$$

$$\mathbb{E}_{\delta_j^{(i)\text{SED}}}[R] = \langle D^2_{\boldsymbol{h}_j^{(i)}}(\mathcal{L} \circ F')[\boldsymbol{h}_j^{(i)}], \frac{\mathbb{E}\delta_j^{(i)}\delta_j^{(i)\top}}{2} \rangle$$

$$= \langle D^2_{\boldsymbol{h}_j^{(i)}}(\mathcal{L} \circ F')[\boldsymbol{h}_j^{(i)}], (\frac{(J^{(i)} - S^{(i)})p}{2J^{(i)}}(\tilde{\boldsymbol{h}}_j^{(i)} - \boldsymbol{h}_j^{(i)})^{\odot 2}$$

$$+ \frac{(J^{(i)} - S^{(i)})(1-p)(J^{(i)} - pJ^{(i)} + pS^{(i)})}{2J^{(i)}S^{(i)}}\boldsymbol{h}_j^{(i)\odot 2})\rangle$$

It is easy to check that the statement satisfies given the value calculated.  □

# B  Implementation Details

## B.1  Missing Algorithm

---

**Algorithm 2** Pipeline of GST+EFD

---

**Require:** A preprocessed training graph dataset $\mathcal{D}_{\text{train}} = \{(\bigoplus \mathcal{G}_j^{(i)}, y^{(i)})\}_{i=1}^n$. A parameterized backbone $F$ and a prediction head $F'$. A historical segment embedding table $\mathcal{T}$.

1: **for** $t = 1$ to $T_0$ **do**
2:     $\mathcal{B} \leftarrow \text{SampleMiniBatch}(\mathcal{D}_{\text{train}})$
3:     **for** $(\mathcal{G}^{(i)}, y^{(i)})$ in $\mathcal{B}$ **do**
4:         $\{\mathcal{G}_s^{(i)}\}_{s \in \mathcal{S}^{(i)}} \leftarrow \text{SampleGraphSegments}(\mathcal{G}^{(i)})$
5:         $\tilde{\boldsymbol{h}}_j^{(i)} \leftarrow \mathcal{T}.\text{LookUp}((i,j))$ **for** $j \notin \mathcal{S}^{(i)}$
6:         $\boldsymbol{h}_s^{(i)} \leftarrow F(\mathcal{G}_s^{(i)})$    **for** $s \in \mathcal{S}^{(i)}$
7:         $\mathcal{T}.\text{InsertOrUpdate}((i,s), \boldsymbol{h}_s^{(i)})$
8:     **end for**
9:     SGD on $\frac{1}{|\mathcal{B}|}\sum_i \mathcal{L}\left(F'(\eta_s^{(i)} \cdot \boldsymbol{h}_s^{(i)} \bigoplus (\eta_j^{(i)} \cdot \tilde{\boldsymbol{h}}_j^{(i)}), y^{(i)}\right)$ {SED with $\eta$ defined in Equation 1}
10: **end for**
11: # Prediction Head Finetuning
12: $\mathcal{T}.\text{InsertOrUpdate}((i,j), F(\mathcal{G}_j^{(i)}))$ for every $\mathcal{G}_j^{(i)}$
13: **for** $t = T_0$ to $T_1$ **do**
14:     **for** $\mathcal{G}^{(i)}$ in SampleMiniBatch($\mathcal{D}_{\text{train}}$) **do**
15:         $\bar{\boldsymbol{h}}_j^{(i)} \leftarrow \mathcal{T}.\text{LookUp}((i,j))$ **for** $j \leq J^{(i)}$
16:     **end for**
17:     SGD with loss $\leftarrow \frac{1}{|\mathcal{B}|}\sum_i \mathcal{L}\left(F'(\bigoplus \bar{\boldsymbol{h}}_j^{(i)})\right)$
18: **end for**

---

Table 4: Overview of the graph datasets used in this study.

| | Avg. # nodes | Min. # nodes | Max. # nodes | Avg. # edges | Min. # edges | Max. # edges |
|---|---|---|---|---|---|---|
| MalNet-Tiny | 1,410 | 5 | 4,994 | 2,860 | 4 | 20,096 |
| MalNet-Large | 47,838 | 3,374 | 541,571 | 225,474 | 20,597 | 3,278,318 |
| TpuGraphs | 38,444 | 299 | 615,019 | 62,475 | 380 | 1,058,278 |

We follow GraphGym [36] to represent design spaces of GNN as (message passing layer type, number of pre-process layers, number of message passing layers, number of post-process layers, activation, aggregation). Our code is implemented in PyTorch [22].

**Implementation details for MalNet-Large.** We consider three model variations for the MalNet-Large dataset. Please refer to their hyperparameters in Table 5. We use Adam optimizer [18] with the base learning rate of 0.01 for GCN and SAGE. For GraphGPS, we use AdamW optimizer [20] with the cosine scheduler and the base learning rate of 0.0005. We use L2 regularization with a weight decay of 1e-4. We train for 600 epochs until convergence. For Prediction Head Finetuning, we finetune for another 100 epochs. We limit the maximum segment size to 5,000 nodes, and use a keep probability $p = 0.5$ if not otherwise specified. We train with CrossEntropy loss.

Table 5: Detailed GNN/Graph Transformer designs used in MalNet-Tiny and MalNet-Large.

| model | GCN | SAGE | GraphGPS |
|---|---|---|---|
| message passing layer type | GCNConv | SAGEConv | GatedGCN+Performer |
| pre-process layer num. | 1 | 1 | 0 |
| message passing layer num. | 2 | 2 | 5 |
| post-process layer num. | 1 | 1 | 3 |
| hidden dimension | 300 | 300 | 64 |
| activation | PReLU | PReLU | ReLU |
| aggregation | mean | mean | mean |

**Implementation details for MalNet-Tiny.** We use the same model architectures/training schedules as in the MalNet-Large dataset. The only difference is that as graphs in MalNet-Tiny have no more than 5000 nodes, so we limit maximum segment size to 500 here.

**Implementation details for TpuGraphs.** We only consider SAGE with configurations (SAGEConv, 0, 4, 3, 128, ReLU, sum) for the TpuGraphs dataset. We use Adam optimizer with the base learning rate of 0.0001. We train for 200,000 iterations until convergence. We by default limit the maximum segment size to 8,192 nodes, and use a keep probability $p = 0.5$ if not otherwise specified. Since we care more about relative ranking than the absolute runtime, we use PairwiseHinge loss within a batch during training:

$$\mathcal{L}(\widehat{y}_1, \widehat{y}_2) = \sum_i \sum_j \mathbb{I}[y_i > y_j] \cdot \max(0, 1 - (\widehat{y}_i - \widehat{y}_j))$$

## C  Additional Results

**Convergence analysis.** To study the effect on convergence for the proposed framework GST and technique SED, we visualize training/test curve per epoch on TpuGraphs dataset in Figure 5, MalNet-Tiny dataset in Figure 6. We show that the convergence speed of various methods studied are quite similar. In addition, the convergence speed in terms of iterations is similar between Full Graph Training and the proposed GST. Due to certain implementation overheads, e.g., on-the-fly graph segment extraction, embedding table query, we didn't observe speed up in terms of wall-clock time for our current implementation yet. Nevertheless, the implementation can be optimized further to reduce the overhead, we leave it for future work.

**Ablation study on partition algorithms.** We incorporated an ablation study focusing on various graph partition algorithms. The test accuracy of GST+EFD using the SAGE backbone for 5 iterations is depicted in Table 6. Our findings indicate that the Random Edge-Cut algorithm delivers subpar results due to its inability to maintain the integrity of the subgraph structure. Some attentive readers might hypothesize that ignoring edges between segments could lead to a decrease in graph property prediction accuracy, leading us to further investigate Vertex-Cut partition algorithms. In theory, Vertex-Cut techniques, which distribute edges across different machines and duplicate nodes as necessary, are likely to result in less information loss compared to Edge-Cut methods. Our empirical findings show that all partition algorithms that manage to retain the local structure have quite similar performance levels. This suggests that edges connecting different segments do not have a significant impact on the final prediction accuracy.

## D  Broader Impact

The paper presents Graph Segment Training (GST), a framework for predicting properties of large graphs using a divide-and-conquer approach. This method addresses the challenge of memory

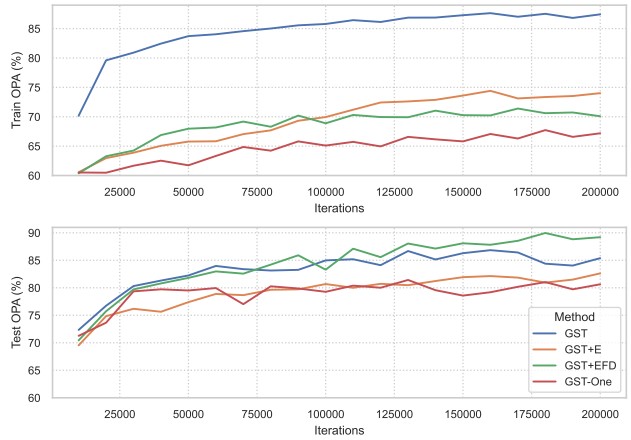

Figure 5: Accuracy curve on TpuGraphs dataset.

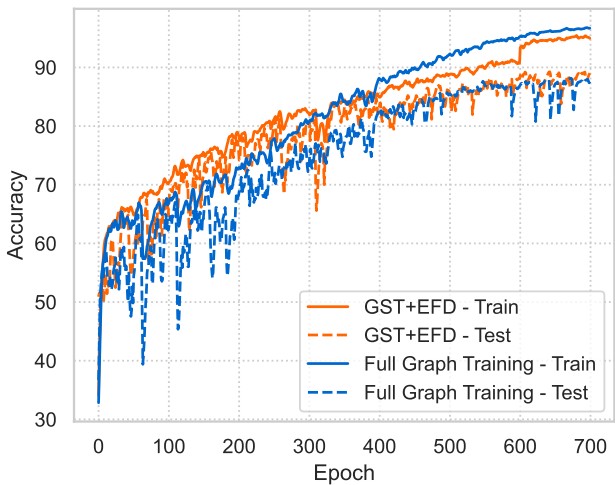

Figure 6: Accuracy curve on MalNet-Tiny dataset.

limitation during training and has the potential to bring significant impact across various domains, with notable efficiency and accuracy improvements. GST can be particularly beneficial to industries that need to manage and interpret large-scale graph data. In telecommunications, it can help optimize network infrastructure, while in cybersecurity, it could improve anomaly detection in network traffic. Furthermore, enabling the capability in large network analysis can contribute to more resilient infrastructure, enhancing the quality of life in many communities. As with any advancement in AI, there's a risk that the benefits of this technology will be unevenly distributed, potentially increasing economic disparity. Companies with access to large amounts of data and the computational resources to analyze it might reap disproportionate benefits, which could further exacerbate the digital divide.

# E Limitations

While Graph Segment Training (GST) represents a significant advancement in graph property prediction, there are several potential limitations to this approach, as highlighted below:

**Segmentation Limitations**: The efficacy of the Graph Segment Training (GST) approach is influenced by the proficiency of the graph partitioning procedure. While our empirical findings indicate that various partitioning algorithms that maintain locality tend to produce satisfactory outcomes, partitions created through random edge-cut methods haven't demonstrated the same level of success.

Table 6: Test accuracy when combined with different partition algorithms.

|  |  | MalNet-Tiny | MalNet-Large |
|---|---|---|---|
| Edge-Cut | Random | 85.43±0.98 | 74.02±2.23 |
| Edge-Cut | Louvain | 88.95±0.67 | 89.16±0.85 |
| Edge-Cut | METIS | 89.24±0.53 | 89.78±0.68 |
| Vertex-Cut | Random | 88.12±1.17 | 87.69±1.51 |
| Vertex-Cut | DBH [33] | 88.79±0.74 | 89.28±0.93 |
| Vertex-Cut | NE [39] | 89.16±0.70 | 89.49±0.87 |

**Historical Embedding Table**: The use of a historical embedding table to obtain embeddings for non-sampled segments introduces additional implementation complexity and potential for errors. If not managed properly, it could lead to inefficient memory usage or even slower the training process.

