# OpenReview forum: "Learning Large Graph Property Prediction via Graph Segment Training"
_NeurIPS.cc/2023/Conference — NeurIPS 2023 poster_

### Official Review · Reviewer_U5zV · 2023-07-01

**Soundness:** 3 good
**Presentation:** 3 good
**Contribution:** 3 good
**Rating:** 5
**Confidence:** 4

**Summary:**

This work uses graph segment training to reduce memory requirements in order to address the scalability concerns of training with large graphs. Additionally, to reduce computation time, historic embeddings for graph segments are stored. These historic embeddings are updated once for all embeddings at the end of training and periodically dropped during training according to the Stale Embedding Dropout to reduce the bias of stale embeddings to the prediction head and loss function.

**Strengths:**

Graph segment training is an intuitive and seemingly successful method for achieving high graph property prediction on large graphs while reducing maximum memory requirements and total runtime.

The study of stale embeddings in this work is particularly important as embedding table lookup have been utilized in prior studies.

[1] Zhang, Shichang, et al. "Motif-driven contrastive learning of graph representations." arXiv preprint arXiv:2012.12533(2020).

[2] Tan, Qiaoyu, Ninghao Liu, and Xia Hu. "Deep representation learning for social network analysis." Frontiers in big Data 2 (2019): 2.


**Weaknesses:**

The experiments only evaluate GST and its several variants. Some evaluation on other large graph training techniques should be included. Although some of these are used for node classification, they can easily be adapted and evaluated for graph property prediction.

[3] Zou, Difan, et al. "Layer-dependent importance sampling for training deep and large graph convolutional networks." Advances in neural information processing systems 32 (2019).
[4] Wei-Lin Chiang, Xuanqing Liu, Si Si, Yang Li, Samy Bengio, and Cho-Jui Hsieh. Cluster-gcn: An efficient algorithm for training deep and large graph convolutional networks. In Proceedings of the 25th ACM SIGKDD International Conference on Knowledge Discovery & Data Mining, pages 257–266, 2019.
[5] Huang, Kezhao, et al. "ReFresh: Reducing Memory Access from Exploiting Stable Historical Embeddings for Graph Neural Network Training." arXiv preprint arXiv:2301.07482 (2023).

In particular, [5] also utilizes historical embeddings for training on large graphs. A comparison with this method specifically would be useful to include.


**Questions:**

When partitioning the graph, what is done with the connecting edge(s) between segments? Is this feature information lost? Would we see performance improvements if this structure information was reincorporated into the model. For example, if each graph was processed twice, using different segmentation sets each time, then the potential loss of structural information can be recovered.

Rather than performing stale embedding dropout, would it be reasonable to simply update the segment embedding table at fixed intervals? Or, update stale embeddings rather than dropout according to the stale embedding keep probability.

In Eq. 2, why is the effect of stale historical embeddings being characterized by the difference between the L(F’(h_s^{(i)}⊕\hat(h)_j^{(i)})) and L(F’(⊕h_j^{(i)}))? It doesn’t seem that the sampled graph segments are being accounted for in the calculation of the second loss. To mitigate differences not caused by stale historic embeddings, would it not be better to concatenate the sampled graph segments in both loss calculations? Perhaps this is a confusion of notation, in which case, j should not be overloaded to represent either any segment in J^{(i)} or any segment in J^{(i)} excluding those in S^{(i)}.

---

> ### Author Rebuttal · Authors · 2023-08-09
>
> We thank the reviewer for the valuable feedback and insightful comments. We appreciate the confirmation that the proposed method is important and our proposed method is successful. We hope we are able to address the review’s concerns, and respectfully ask to consider increasing the score.
>
> **Q1. Some evaluation on other large graph training techniques should be included. Although some of these are used for node classification, they can easily be adapted and evaluated for graph property prediction.**
>
> We thank the reviewer for the suggestion. We will integrate the mentioned references into the related work section and provide a discussion. It's worth mentioning that we did assess a baseline, GST-One, that trains on a singular sampled segment for each cycle and, when contextualized, shares significant similarities with Cluster-GCN [4]. Additionally, we'd like to emphasize that adapting prior methods, such as [5] which uses historical embeddings, to our scenario is impractical. The method in [5] retains a historical embedding for every node, which makes it unfeasible to store all the historical embeddings given our conditions, i.e., for the malnet-large dataset, the embedding table needs to store approximately $2 \times 10^9$ rows. Taking into account the hidden dimension, this amounts to about 1TB of memory. The memory requirement would further surge if we raise the count of training graphs.
>
> [5] Huang, Kezhao, et al. "ReFresh: Reducing Memory Access from Exploiting Stable Historical Embeddings for Graph Neural Network Training." arXiv preprint arXiv:2301.07482 (2023).
>
> **Q2. When partitioning the graph, what is done with the connecting edge(s) between segments? Is this feature information lost? Would we see performance improvements if this structure information was reincorporated into the model.**
>
> We thank the reviewer for the question. There's a possibility of some information loss, but our empirical studies show it doesn't greatly impact performance. Addressing the reviewer's query, we delve deeper into Vertex-Cut partition algorithms in Table 5 of Appendix. Theoretically, Vertex-Cut approaches, which distribute edges among various machines and replicate nodes as required, might experience less information loss compared to Edge-Cut techniques. Our hands-on results indicate that all partitioning algorithms which preserve local structure exhibit comparable performance. This infers that edges linking different segments don't substantially influence the ultimate prediction accuracy.
>
> **Q3. Rather than performing stale embedding dropout, would it be reasonable to simply update the segment embedding table at fixed intervals? Or, update stale embeddings rather than dropout according to the stale embedding keep probability.**
>
> We thank the reviewer for the comment. While it is feasible to proceed in that manner, it's important to highlight that refreshing the entire segment embedding table might require more time than completing one training epoch (as the number of segments is usually an order larger than the number of segments we trained on every iteration), resulting in significant overhead. Furthermore, our results indicate that GST+EFD outperforms GST in accuracy. Hence, we believe it's improbable for this alternative to surpass our suggested algorithm in either efficiency or accuracy.
>
> **Q4. In Eq. 2, why is the effect of stale historical embeddings being characterized by the difference between the L(F’(h_s^{(i)}⊕\hat(h)_j^{(i)})) and L(F’(⊕h_j^{(i)}))? It doesn’t seem that the sampled graph segments are being accounted for in the calculation of the second loss. To mitigate differences not caused by stale historic embeddings, would it not be better to concatenate the sampled graph segments in both loss calculations? Perhaps this is a confusion of notation, in which case, j should not be overloaded to represent either any segment in J^{(i)} or any segment in J^{(i)} excluding those in S^{(i)}.**
>
> We thank the reviewer for the comment. Indeed, the reviewer has a valid point. We reloaded $j$ to save some space in notation. We will try the reviewer’s recommendation in the final version.

---

> > ### Comment · Reviewer_U5zV · 2023-08-16
> >
> > The authors have successfully addressed most of my concerns. The main contribution of this work is realizing GNN training with very large graphs on limited resources. The proposed algorithm is simple and effective. However, such type of solutions would require a ton of technical tricks and engineering effort for arbitrary large graph data. Fortunately the authors should have implemented those on the set of data they investigated. Open-sourcing the solutions, no matter where the paper is accepted, would be the critical factor of the impact and contribution of the work. Generally describing the segment-then-train idea and reporting a set of numbers from the authors' own side would not help the community significantly. I'll raise my score. I hope the authors would consider seriously about the real impact of the work.

---

> > > ### Author Response · Authors · 2023-08-17
> > > **Reply by Authors**
> > >
> > > Thank you for your insightful review and for recognizing the value of our work on GNN training with large graphs. We appreciate your emphasis on the importance of open-sourcing our solutions. In response to your comments, we are pleased to inform you that we have already released the code. We believe this step aligns with your suggestions and will contribute positively to the community.

---

### Official Review · Reviewer_Uc8u · 2023-07-03

**Soundness:** 4 excellent
**Presentation:** 4 excellent
**Contribution:** 4 excellent
**Rating:** 8
**Confidence:** 4

**Summary:**

This paper aims to predict properties of very large graphs, by segmenting the large graph into multiple subgraphs with the existing graph partitioning algorithm and then learning over segmented subgraphs where gradients are calculated on some of them for memory-efficient training. Also, to further efficiently train with segmented subgraphs, the authors use the embedding table that stores subgraph representations and provides embeddings for subgraphs that are not back-propagated, while calculating embeddings for remaining subgraphs. Moreover, in order to prevent the staleness issue where subgraph representations in the embedding table are outdated, the authors not only finetune only the property prediction head as the post step of training but also randomly drop some subgraph representations with regard to their stalenesses. The authors validate the proposed method, namely Graph Segment Training (GST), with its variants (GST+EFD) on multiple large-scale graph datasets, showing the efficacy of the proposed methods.

**Strengths:**

* The idea of training with partitioned subgraphs of the large graph for its property prediction problem is interesting, novel, and highly valuable to the graph community.
* Each proposed ingredient (e.g., historical embedding table, prediction head fine-tuning, and stale embedding dropout), which composes the final GST+EFD architecture, has its own unique benefit in learning with segmented subgraphs of the large graph; having solid contributions.
* The theoretical results show the benefit of stale embedding dropout, which randomly drops some subgraphs with respect to their stalenesses and results in reducing bias from stale embeddings.
* The proposed GST and GST+EFD outperform the full graph training mechanism while being much more efficient.
* This paper is extremely well-written. All contents are clear and easy to follow.

**Weaknesses:**

* I don't see any.

**Questions:**

* How to fetch the embeddings of subgraphs from the embedding table, if the embedding table does not have the representations for them. For example, at the beginning of training, the historical embedding table is empty and there may be no representations of partitioned subgraphs.
* It is unclear why GST+EFD can outperform GST, given that GST calculates all subgraph embeddings every time, while GST+EFD sometimes uses the stable subgraph embeddings from the embedding table.

**Limitations:**

The authors do not discuss the limitations and potential negative societal impact of their work.

---

> ### Author Rebuttal · Authors · 2023-08-09
>
> We thank the reviewer for the valuable feedback and insightful comments. We appreciate the reviewer for confirming that our paper is novel, extremely well-written and highly valuable in the graph community.
>
> **Q1. How to fetch the embeddings of subgraphs from the embedding table, if the embedding table does not have the representations for them. For example, at the beginning of training, the historical embedding table is empty and there may be no representations of partitioned subgraphs.**
>
> We thank the reviewer for the question. For historical embeddings, we employ a 0 initialization. Therefore, at the onset of training, if a segment hasn't been updated previously, its representation defaults to a vector of zeroes.
>
> **Q2. It is unclear why GST+EFD can outperform GST, given that GST calculates all subgraph embeddings every time, while GST+EFD sometimes uses the stable subgraph embeddings from the embedding table.**
>
> We appreciate the reviewer's feedback. Our presumption is that the introduced Staled Embedding Dropout offers supplementary regularization, fostering improved feature representation. Concurrently, Prediction Head Finetuning assists in learning alignment and thus mitigates staleness.

---

> > ### Comment · Reviewer_Uc8u · 2023-08-16
> >
> > Thank you for addressing my questions. After reading other reviews and responses, I do not have any more concerns or questions.

---

> > > ### Author Response · Authors · 2023-08-16
> > > **Reply by Authors**
> > >
> > > We appreciate your feedback and are pleased to acknowledge that our responses have successfully addressed your concerns!

---

### Official Review · Reviewer_vMFq · 2023-07-06

**Soundness:** 3 good
**Presentation:** 3 good
**Contribution:** 3 good
**Rating:** 5
**Confidence:** 3

**Summary:**

This paper studies an important problem on large graphs.
The authors propose a new Graph Segment Training (GST) method for large-scale prediction of properties. The proposed method utilizes a divide-and-conquer approach to allow learning large graph property prediction with a constant memory footprint. GST divides a large graph into segments and then backpropagates through a few segments sampled per training iteration.
Extensive experiments demonstrate the effectiveness of proposed method.

**Strengths:**

- This work addresses an important problem of property prediction on large graphs, which has applicability in many real-world settings.
- The proposed GST framework uses a divide-and-conquer approach to enable large-scale property prediction.
- The experiments are well-designed and demonstrate the effectiveness of the proposed framework.

**Weaknesses:**

- The paper could benefit from a more detailed discussion of the limitations and potential future directions of the proposed framework.


**Questions:**

- How efficiency does the proposed framework compare to existing methods for large graph property prediction?
- Could the proposed method be used to other types of large graphs, such as social networks or biological networks?
- How sensitive is the proposed framework to hyperparameters?

**Limitations:**

The paper could benefit from a more comprehensive discussion of the limitations and broader societal impact of the proposed framework.

---

> ### Author Rebuttal · Authors · 2023-08-09
>
> We thank the reviewer for the valuable feedback and insightful comments. We appreciate the confirmation that the problem we studied is important and experimental design is well-conducted. We hope we are able to address the review’s concerns, and respectfully ask to consider increasing the score.
>
> **Q1. How efficiency does the proposed framework compare to existing methods for large graph property prediction?**
>
> We thank the reviewer for the question. Our GST framework is tailored to train on large graphs, a task that previous methods struggled with due to memory limitations. This highlights our method's memory efficiency in contrast to earlier approaches. As depicted in Table 3, our GST framework also boasts improved runtime efficiency (3x faster). Importantly, our final algorithm, GST+EFD, introduces only a slight increase in time compared to GST-One. The latter is trained on a singular sampled segment per cycle and can be viewed as a theoretical minimum time benchmark.
>
> **Q2. Could the proposed method be used for other types of large graphs, such as social networks or biological networks?**
>
> We appreciate the reviewer's question. Indeed, our suggested approach is not limited to any particular type of graph. Most existing social network datasets tend to emphasize node-level or link-level tasks. On the other hand, the publicly available biological networks are generally of a smaller scale. This is the reason we didn’t test on these two types of graphs.
>
> **Q3. How sensitive is the proposed framework to hyperparameters?**
>
> We thank the reviewer for the question. We conducted an ablation study adjusting some design parameters as shown in Figures 2-4. Our findings indicate that the GST+EFD configuration is notably robust to variations in maximum segment size and partition algorithms. It's worth noting that our optimization parameters remain consistent across all methods and are adopted directly from earlier implementations for Full Graph Training.

---

> > ### Comment · Reviewer_vMFq · 2023-08-18
> >
> > Thanks for your response, I will keep my score.

---

### Official Review · Reviewer_rf6m · 2023-07-07

**Soundness:** 3 good
**Presentation:** 3 good
**Contribution:** 3 good
**Rating:** 6
**Confidence:** 4

**Summary:**

This paper deals with large graph learning tasks via graph segmentations. More specifically, in each training step, the author sample nodes from graph segmentations and only update parameters related to the selected nodes. To optimize memory consumption, the author further introduced a historical embedding table. To bridge the training and prediction gap, the author also designed a prediction head fine-tuning scheme. Besides, the author provided some theoretical analysis of the proposed method. Empirical results show the proposed method remains good memory efficiency and test accuracy.

**Strengths:**

1. The paper targets an important problem with high application value, which makes property predictions on Large graphs. The proposed segment training idea looks reasonable and works well practically.

2. Based on the graph segmentation idea, the author has made comprehensive and detailed consideration of the memory usage, the training-testing gap, and the theoretical analysis of the historical approximation bias, which makes the proposed method more technically sound.

3. The paper is well-organized and easy to follow.

**Weaknesses:**

1. The proposed method didn't discuss how to deal with the inter-segmentation training, for example, how to learn a model with link prediction between two nodes from different graph segmentations.

2. About Baselines: The experimental part does not compare with other large-graph learning baselines such as GraphSage[1]. More baselines should be compared to evaluate the effectiveness of proposed GST.

Reference:
[1] GraphSage: Representation Learning on Large Graphs, NIPS 2017.

**Questions:**

1. How to adapt the proposed method with graph properties learning that are between segmentations?

**Limitations:**

The proposed method can process large-scale graphs, which might be used for user information mining on large social networks. There are potential risks to user privacy when applying such methods.

---

> ### Author Rebuttal · Authors · 2023-08-09
>
> We thank the reviewer for the valuable feedback and insightful comments. We appreciate the reviewer for confirming that our paper is technically sound and easy to follow. We respectfully ask the reviewer to consider increasing the score if our clarification has addressed the concerns raised by the reviewer.
>
> **Q1. How to learn a model with link prediction between two nodes from different graph segmentations.**
>
> We thank the reviewer for the question. We would like to clarify that we use “Graph property prediction” to denote that the entire graph should deserve one prediction (contrast with: node-level, and edge-level, respectively, where each node and edge should receive a prediction). “Graph property prediction” includes “graph classification” (both single label or multi-label setting), “regression” (e.g., in chemical molecules, a graph-level GNN could output continuous values such as “boiling temperature”), ranking, and so on. Explicitly trying to predict properties of a given link connecting two segmentations is essentially a link prediction problem, which we believe is not in the scope of this paper.
>
> **Q2. The experimental part does not compare with other large-graph learning baselines such as GraphSage.**
>
> We thank the reviewer for the comment. Most of the prior research addressing either node-level or link-level prediction issues fails to directly translate to large-scale graph property prediction tasks. Nevertheless, we did evaluate one baseline - GST-One, in our tests. It operates by training on a single sampled segment per cycle, which, if we adjust to this context, bears notable resemblance to either GraphSAGE or Cluster-GCN. Our findings indicate that this approach resulted in subpar performance.

---

### Decision · Program_Chairs · 2023-09-21

**Decision:**

Accept (poster)

**Comment:**

The reviewers in general give positive ratings to this paper. The author rebuttal answered many questions asked in the reviewers. The remaining weaknesses pointed by the reviewers include: (1) the lack of a more detailed discussion of the limitations and potential future directions of the proposed framework; (2) limited experimental evaluations, with missing large graph training techniques that should be included.

I am happy to recommend this paper as accept. I look forward to seeing the authors addressed the aforementioned remaining issues in the camera-ready submission.